# Factors Associated with the Prevalence of Postpartum Anxiety Disorder and Depression in Syrian Migrant Women Living in Turkey: A Cross-Sectional Study

**DOI:** 10.3390/healthcare11182517

**Published:** 2023-09-12

**Authors:** Muhammed Atak, Mehmet Akif Sezerol, Mehmet Sait Değer, Hamza Kurubal

**Affiliations:** 1Department of Public Health, Istanbul Medical Faculty, Istanbul University, Istanbul 34093, Türkiye; 2Epidemiology Program, Institute of Health Sciences, İstanbul Medipol University, Istanbul 34810, Türkiye; masezerol@gmail.com; 3Health Management Program, Graduate Education Institute, Maltepe University, Istanbul 34857, Türkiye; 4Department of Public Health, School of Medicine, Istanbul Medipol University, Istanbul 34810, Türkiye; 5Sultanbeyli District Health Directorate, Istanbul 34935, Türkiye; 6Department of Public Health, Medical Faculty, Hitit University, Corum 19030, Türkiye; mehmetsaitdeger@hitit.edu.tr; 7Klinik Porta Westfalica GmbH & Co. KG, Steinstrasse 65, 32547 Bad Oeynhausen, Germany; h.kurubal@kpw.eu

**Keywords:** immigrant, postpartum, postpartum anxiety, postpartum depression, maternal health, perinatal healthcare

## Abstract

The global migration trend has brought attention to the mental health of immigrant populations, especially postpartum women. The prevalence of postpartum anxiety and depression among these populations remains a growing concern. This study aimed to explore the factors associated with postpartum anxiety and depression among immigrant women, with a special emphasis on Syrian migrants in Turkey. A cross-sectional design was employed, enrolling postpartum women who visited the Strengthened Migrant Health Center in Istanbul between July and December 2022. Data were collected using a three-part questionnaire comprising sociodemographic details and scores from the Edinburgh Postpartum Depression Scale (EPDS) and Beck Anxiety Scale (BAI). The average age of participants was 25.73 years. The mean EPDS and BAI scores were low at 1.27 and 3.85, respectively. Notably, 97.1% of women scored below the EPDS cut-off point of 13. EPDS scores showed a significant relationship between income levels and COVID-19 vaccination status, while BAI scores were significantly associated with educational levels. There was an inverse correlation between EPDS scores and the number of cohabitants in a household. The observed low rates of postpartum depression and anxiety could be attributed to the accessible health services and psychosocial support for immigrants in Turkey. It would be useful to conduct multicenter and comprehensive epidemiological studies on migrant puerperas.

## 1. Introduction

Pregnancy, although a natural occurrence, introduces profound physical, emotional, and psychological changes that can have adverse impacts on mothers [1,2]. Research indicates that both pregnancy and the postpartum period can elevate the risk of mild depressive and anxiety symptoms [3,4].

Globally, postpartum anxiety and depression are estimated by the World Health Organization (WHO) to affect 13% of women, with rates varying between 9.5% and 25.8% [5]. Remarkably, this prevalence is more pronounced among immigrant and refugee women [6,7]. It is suggested that immigrant women face anxiety and depression levels 2–5 times greater than the general population [8,9]. Research attributes this increased vulnerability in immigrant women to various challenges faced during the migration process and after settling, with reported postpartum anxiety and depression rates among them ranging from 20% to 50% [2,10].

Beyond the psychological toll on the mother, postpartum anxiety and depression can detrimentally influence the mother–infant bond as well as the infant’s cognitive, emotional, and behavioral development [11,12]. Several psychosocial determinants, including genetic predisposition, hormonal fluctuations during and after pregnancy, socioeconomic status, marital dynamics, traumatic experiences, and insufficient social support, have been associated with postpartum depression [5,7,11,13,14]. Alarmingly, mental health issues among migrant women often remain underrecognized [15].

The WHO identifies postpartum anxiety and depression as an escalating public health concern and underscores maternal mental health as a pivotal determinant of both maternal and infant health outcomes [2,16,17]. As highlighted in the International Organization for Migration (IOM) 2022 report, there are currently 89.4 million displaced individuals worldwide, with women constituting approximately 48% of this demographic [10,18]. In Turkey alone reside around 3.5 million Syrian refugees, of whom 46.5% are female [19]. These migrant women, facing compounded traumas from migration and multifaceted challenges in their new environments, are particularly susceptible to psychological disorders, which are further exacerbated during pregnancy [10].

Concerningly, there is a notable surge in psychiatric hospital admissions among women following childbirth, persisting for up to two years in certain instances [6,10]. Timely recognition and appropriate intervention for postpartum anxiety and depression are pivotal, especially in the context of migrants’ healthcare accessibility and utilization [20,21]. Therefore, there is an imperative need to integrate epidemiological research on postpartum anxiety and depression risk factors within primary healthcare services and to champion inclusive public health strategies [4,22,23]. Given these considerations, this study aims to assess the prevalence of postpartum anxiety and depression, along with their associated factors, among Syrian migrant women attending a Migrant Health Center in Istanbul.

## 2. Materials and Methods

### 2.1. Study Design and Participants

This research was planned as a cross-sectional study. Strengthened Migrant Health Centers are organizations that provide primary healthcare services to Syrian refugees who have settled in Turkey. These centers include specialist physicians, general practitioners, dentists, assistant health personnel, psychologists, and social workers. The majority of the workers in the centers are Syrian health workers. Therefore, there is no language barrier. Eight of these centers are located in Istanbul. The population of the research consists of Syrian postpartum women who applied to a Strengthened Migrant Health Center located in Istanbul. Postpartum women apply to the centers for health check-ups at least four times during their pregnancy and at least three times in the first 42 days after delivery. The research was conducted between the dates of 15 July 2022 and 31 December 2022. Syrian women who applied to the migrant health center and were within the first 6 weeks after delivery were included in the study. All postpartum women who applied to the center without sampling were included in the study. Migrant women who refused to participate in the study and who were more than 6 weeks postpartum were excluded.

### 2.2. Data Collection Tool and Measurements

In the study, a questionnaire consisting of three parts was administered face-to-face to postpartum women who applied to the center. There are 62 questions in total in the entire questionnaire.

### 2.3. Definitive Measurements

The first part of the questionnaire consists of 31 questions. In this section, participants were asked questions about their age, income status, education level, number of children, number of people in the household, chronic disease status, smoking and alcohol use, COVID-19, pregnancy follow-ups, pregnancy support treatments, baby’s health status, and baby-related health screenings.

### 2.4. Depression Measurement

The Edinburgh Postpartum Depression Scale, which has been used in many studies, was used to measure the depression level of postpartum Syrian women. EPDS is a scale used to assess postpartum depressive symptoms and consists of 10 items. This scale includes statements about how the mother felt during the last week [24]. Items were designed in a 4-point Likert format. The maximum score that can be taken from the scale is 30, and the minimum is 0. The level of depression increases as the score increases. Thirteen points were accepted as the cut-off level for postpartum depression [25]. Those who scored 13 points or more on the scale were considered to be at risk for depression.

### 2.5. Anxiety Measurement

The Beck Anxiety Inventory was used to measure the anxiety level of Syrian postpartum women. The Beck Anxiety Scale consists of 21 items. Items were designed in a 4-point Likert format. The BAI is a validated, self-administered inventory that measures anxiety symptoms over the previous 7 days. With reference to the studies conducted by Beck and Dantas, the scores obtained from the scale were evaluated in 3 groups in terms of anxiety: ˂20 points were considered no risk or low risk, 20–30 points as medium risk, and 31 points and above as high risk [26,27].

### 2.6. Statistical Analysis

In the analysis of the data obtained in the study, categorical variables were presented as numbers and percentages, and continuous variables were presented as mean and standard deviation. For the normality analysis of the data, Kolmogorov–Smirnov and Shapiro–Wilk tests were performed, and Skewness and Kurtosis values of the scales with *p* < 0.05 were examined. The values with Skewness and Kurtosis values between ±2 were accepted to be normally distributed, and parametric tests were applied. Chi-square and Fisher’s exact tests were used to compare categorical variables between groups. Student t-tests and one-way ANOVA tests were used for the statistical analysis of normally distributed data. The Pearson chi-square test was used for the relationship between continuous variables. In the statistical analysis, *p* ˂ 0.05 was considered significant. The statistical analysis of the research data was conducted with the SPSS 25.0 package program.

### 2.7. Ethic Decision

This study was conducted in strict adherence to the guidelines set out in the Helsinki Declaration. Ethical approval was granted by the Ethics Committee for Non-Interventional Clinical Studies at Istanbul Medipol University, as confirmed by the decision dated 6 July 2022 and numbered 604. Informed consent was obtained from all participants after they were informed about the research and permissions.

## 3. Results

A total of 172 migrant postpartum women participated in the study. The mean age and standard deviation (SD) of the participants were 25.73 ± 5.9, the youngest age was 16.37, and the oldest was 43.99. Concerning the participants, 70.3% were in primary school or below, and 92.0% of them stated that they had a pregnancy follow-up.

In the study, 7% (12 people) of the participants stated that they have a chronic disease. The 12 people with chronic diseases stated that their diseases were hypertension (1 person), diabetes (2 people), thyroid diseases (4 people), other diseases (4 people), and 1 person had AIDS.

Furthermore, 95.9% (165 people) of the research group gave birth at a normal gestational week (37–42 weeks), 96.5% (166 people) stated that their child was born in the normal delivery range (2500–4000 g), and 98.3% (169 people) stated that their baby did not have any health problems.

Furthermore, 98% (168 people) of the postpartum women in the research group stated that they took folic acid supplementation during pregnancy and still received iron supplementation with D-vit. All participants stated that they knew that they should breastfeed their children until they were 6 months old and that they should feed their babies breast milk with additional foods after 6 months until at least 2 years of age.

Health evaluation status is a subjective description of an individual’s assessment of their health status. Almost all migrant women in the research group (99.4%) stated that they perceived their health status as good or very good.

The data on the sociodemographic characteristics of the participants and their use of health services during pregnancy and postpartum are shown in Table 1.

The Edinburgh Postpartum Depression Scale (EPDS) score and SD values of the participants in the study were determined as 1.27 ± 3.72. In the evaluation made according to the EPDS cut-off point (13), 97.1% of the puerperal women got scores below the cut-off point (13) and 2.9% of them got scores above the cut-off point value. Beck Anxiety Inventory (BAI) mean and SD values were determined as 3.85 ± 2.62. All of the participants had scores of 21 and below. Moderate and severe anxiety were not observed in the participants (Table 2).

According to the results of the chi-square relationship test performed to determine the relationship between the participants’ EPDS score and their income level, a statistically significant relationship was found between income level and EPDS score (*p* 0.05). Depression scores were higher in those whose income level was below the minimum wage (5.324 TL) (89.8%). According to the results of the chi-square relationship test performed to determine the relationship between the EPDS scores of the participants and their COVID-19 vaccine status, a statistically significant relationship was found between the COVID-19 vaccine status and the EPDS score (*p* 0.05). The rate of depression was found to be lower in those who did not get vaccinated than in those who did. (Table 3).

A statistically significant relationship was found between the Beck Anxiety Inventory (BAI) score and education level (*p* 0.05). Anxiety scores of those with primary education (62.5%) and high school (25%) education levels were found to be higher than those with less than primary education and associate-bachelor degrees. (Table 4).

A negative correlation (cc −0.153) was found between the EPDS score and the number of people living together at home, which is one of the independent variables (*p* 0.05). As the number of people living together increases, the postpartum depression situation decreases.

## 4. Discussion

According to the data of the International Organization for Migration (IOM), globally, 89.4 million people are displaced (refugees, asylum seekers, or exposed to internal migration) [18]. The average age of mothers in Turkey is 29 years (2020) [28]. The average maternal age of migrant women in the research group is 25.7 years, which is lower than the average age in Turkey. In the study group, the proportion of those in the risky pregnancy category (over 35 years of age) was 10.5%. This suggests that women in the high-risk age group do not use family planning services sufficiently or have problems accessing these services. However, with the participation of women in working life at our age, the gestational age is gradually increasing. Gestational age and mean gestational age values should be evaluated in each group by taking into account social, environmental, and psychosocial factors. The idea of continuing the lineage, education level, and socio-cultural values affect the pregnancy preferences and healthcare use of immigrant women [29].

The vast majority (91.9%) of the migrant women in the research group had regular pregnancy follow-ups, and ¾ (75.2%) of them had their follow-ups done in public health facilities. Furthermore, 44.6% of them had their follow-up from a gynecologist at a state hospital, and it is seen that immigrant women generally access and receive health services. The fact that almost all (98%) of the migrant women in the study group take folic acid, iron, and Devit supplements during pregnancy suggests that they have knowledge and attitudes about healthy pregnancy. Studies have shown that the prenatal care of migrant women is inadequate [30]. Again, in scientific studies conducted in different countries, it has been shown that migrant women and ethnic groups that are disadvantaged in extraordinary situations such as pandemics have more difficult access to health services than other groups in society [31,32].

Immunization is a key component of primary healthcare, which plays a critical role in the prevention of infectious diseases and epidemics [33]. Pregnancy tetanus vaccination, which prevents tetanus-related neonatal deaths very seriously and is still the most effective method of eliminating maternal and neonatal tetanus, is at a vital point, and it is recommended to be administered at least two doses during pregnancy [34,35,36]. In a study conducted in Turkey, the rate of Syrian migrant women who had two doses of tetanus vaccine during pregnancy was reported as 29.7% [37]. The rate of tetanus vaccination during pregnancy among Turkish women is 81%. Although the majority (91.9%) of the migrant women in the study group had regular follow-ups during pregnancy, the tetanus vaccination rate during pregnancy was determined to be 20.3%. In addition, the majority (58.1%) of the immigrant women in the research group stated that they had never had a tetanus vaccination before, which is a very high rate. Every pregnant woman should be vaccinated against diseases like tetanus, which poses a high risk of mortality for both the mother and the neonate. This is an important issue that needs to be addressed and examined. The data showing that the vast majority of migrant women in our research group have access to health services can be seen in this study. In this respect, more comprehensive studies on pregnancy vaccination among immigrant women are needed. In addition, new methods should be developed to increase pregnancy vaccination rates by conducting studies for those who are hesitant about vaccination.

It is seen that the vast majority (90.1%) of the migrant women in the research group had their post-pregnancy follow-ups done at the family health center or migrant health center. Almost all of the immigrant mothers had their children screened as newborns (99.4%). These data show that immigrant women, in particular, have easy access to health services for all immigrants. A healthy transition from pregnancy to motherhood is of critical importance for the physical, mental, and social well-being of both mother and baby [38]. In a study conducted with Syrian immigrants in Jordan, it was stated that biological and socio-cultural factors negatively affect the mental and social well-being of individuals, and in this sense, immigrants should be supported [20]. Identification of women at risk for anxiety and depression during pregnancy and early puerperium provides an opportunity to improve the health status of women and children [39]. The fact that primary healthcare workers provide health services with a more sensitive approach to disadvantaged immigrants both facilitates the detection of postpartum anxiety and depression and increases the effectiveness of programs that will protect and support immigrant women from postpartum depression [15].

Being infected with COVID-19 during pregnancy increases the risk of premature birth or death [40].

Factors such as the absence of pregnant women in clinical trials of COVID-19 vaccines in the first place, the development of vaccines using new technologies, the inability to clearly demonstrate the efficacy and safety of vaccines during pregnancy and postpartum, and the side effects of vaccines may have been effective in delaying the vaccination preferences of pregnant and postpartum women [41,42,43,44]. In a study conducted with migrant pregnant women in Turkey, it was reported that only 20% of women had two vaccine doses and that vaccination was very low among migrant pregnant women [37]. In another study conducted in Lebanon, the COVID-19 vaccination rate of pregnant Syrians was found to be 25.9% [41]. In another study conducted in Lebanon, similar findings were obtained, indicating that vaccination rates among migrants were lower than those among the host population [45]. In a study in England, less than 1/3 of immigrants and pregnant women were vaccinated, and it was shown that socioeconomic level and being at a young age are associated with vaccination [46]. Although 89.5% of the immigrant women in the research group did not have the COVID-19 disease, it is a positive situation that 45.9% of them had at least one dose of the COVID-19 vaccine. However, more than half of the research group (54.1%) had never been vaccinated, and the number of those who were fully vaccinated remained at a very low rate (<1%). The presence of individuals in the study group who had the disease but were not diagnosed may have affected the vaccination status. This situation causes both individual and social weakness against COVID-19 and allows the spread of the disease. Studies have shown that supportive public health policies and the advice of physicians and midwives are important for the vaccination of immigrants [37,47,48]. In order to achieve success in vaccination, which is the most effective method of combating a globally effective epidemic such as COVID-19 (preventing disease and death), it is undoubtedly very important to vaccinate individuals from all layers of society, including immigrants and asylum seekers [49].

Difficulties such as adapting to a different culture, language, accommodation, education, work, and access to health services due to migration increase the psychosocial effects of immigrant women [9,50]. Factors such as limited language proficiency, low educational and economic levels, ignorance of the health system, and a lack of guiding and informative mechanisms may cause immigrants to not be able to benefit from health services adequately [51]. The fact that almost all (99.4%) of the migrant women in the research group stated that they perceived their health status as good or very good is remarkable considering their current accommodation, nutrition, income status, and psychosocial influences. The fact that immigrant women adapt to their low living standards and develop social and psychological support mechanisms (family, relatives, and society) may have led to their better perceptions of health [16]. In support of this situation, it has been shown that the anxiety and depression levels of mothers who received pregnancy training during pregnancy were low [52].

Immigrant women constitute 48% of the world’s immigrants, and they are more vulnerable to psychological disorders due to their different needs during migration and pregnancy [10,53]. In a pilot study conducted in Lebanon, Syrian migrant women were found to show more depression symptoms than Lebanese women [23]. In a systematic review made in Turkey, it was shown that factors such as low education and income level, insufficient prenatal and antenatal care, low social support, low maternal age, history of depression, and unwanted pregnancy pose risks for anxiety and depression [12]. Furthermore, 97.1% of the migrant postpartum women in the study group were not considered to be at risk for depression, according to the EPDS. Only a very small portion (2.9%) was considered to be at risk for postpartum depression. The fact that the depression level of immigrant postpartum women with low income is higher is proof that the economic situation is effective on mental health. The fact that the depression rates of immigrant postpartum women who did not receive the COVID-19 vaccine were lower than those who were vaccinated may be related to the fact that individuals tend to be vaccinated for fear of contracting the disease. Again, in the evaluation of anxiety in our research group, the BAI scores of all migrant puerperas were ˂20. This value indicates that the puerperium is in the risk-free and lowest-risk group in terms of anxiety. The anxiety level of those with primary and high school education levels was higher than that of those with other education levels. In the literature, it has been shown that as the education level increases, anxiety and depression decrease [54,55,56]. The decrease in the rate of depression with the increase in the number of people living together at home and the presence of psychosocial support environments and mechanisms prevent anxiety and depression. Again, access to health services and the presence of psychosocial support activities are important in coping with mental problems in pregnant and postpartum women [57]. Studies have shown that postpartum care and psychosocial support services for immigrant women are of critical importance and that new public policies should be developed for these services, which are generally not sufficient [16,58]. In addition, it is known that family support has a positive effect on mental health in immigrant women [59].

### Strengths and Limitations

There are many studies in the literature that include psychological evaluations of the postpartum period. In addition, there is a lot of mental health research on immigrant women. However, a postpartum period mental health study among immigrant women has not been found in the literature. In this respect, the most powerful aspect of the study is that it is the first mental health research conducted on Syrian postpartum immigrants.

On the other hand, there are some limitations to our research. The study was conducted at a single center, and the number of participants was relatively small. In addition, the population of the study consisted of those who applied to the health institution. Therefore, it may not be correct to generalize the results obtained for anxiety and depression to all immigrant women. Therefore, it will be beneficial to conduct multicenter and higher participation studies in the future.

## 5. Conclusions

In our study, anxiety and depression levels in the migrant puerperium were found to be low. Factors such as income status, education level, and utilization of health services affected the anxiety and depression levels of migrant puerperium. Migration and immigration are difficult and important concepts that need to be tackled individually, socially, and globally. As a matter of fact, in the world we live in, natural or man-made disasters or crises are experienced every day, and the number of immigrants is increasing day by day. For this reason, careful monitoring and follow-up of immigrants from disadvantaged groups who continue their lives as a part of society are important for public health. Immigrants’ access to health services should be facilitated, and healthcare policies should be revised by making determinations regarding their needs. In this context, special attention should be given to immigrant women, especially during pregnancy and puerperium periods. Inclusive public health efforts and policies should be increased by expanding epidemiological research on postpartum anxiety and depression risk factors and psychosocial support services programs integrated into primary healthcare services for immigrants. Protecting the health and well-being of immigrants with strengthened public health policies will also contribute to the protection of public health.

## Figures and Tables

**Table 1 healthcare-11-02517-t001:** Sociodemographic characteristics of the participants and their use of healthcare services.

Characteristics	Number (n)	Percentage (%)
**Age Group**		
˂18 age	8	4.7
18–25 age	93	54.1
26–34 age	53	30.8
>35 ages	18	10.5
**Educational Level**		
Under Primary Education	14	8.1
Primary Education	107	62.2
High School	43	25
Associate-Bachelor	8	4.7
**Income Level**		
1–5300 * TL	154	89.5
5300–17,000 ** TL	18	10.5
17,001 TL and more	-	-
**Number of people living together**		
2–3 people	46	26.7
4–5 people	57	33.2
6–8 people	51	29.7
9 people and more	17	10.3
**Child Number**		
1 child	55	32
2 children	40	23.3
3 children	38	22.1
4 children and more	38	22.1
**Obstetrics (Regular)**		
Yes	158	91.9
No	14	8.1
**Obstetrics Center**		
Family Health Center	9	4.9
Immigrant Health Center	45	26.1
State Hospital	76	44.2
Private Hospital	37	21.8
Not Vaccinated	5	3
Pregnancy Tetanus Vaccine Status		
Have never done before	100	58.1
Had it done during pregnancy	35	20.3
Not done during pregnancy	37	21.6
**Postpartum Tracking Status (ASM or GSM)**		
Followed up	155	90.1
No follow up	17	9.9
**Baby screening and vaccination status**		
Newborn screening (NTP)	171	99.4
Hearing Scan	164	95.5
**Vaccine (in accordance with the Routine Immunization Schedule)**	172	100
**Having COVID-19 virus**		
Yes	18	10.5
No	154	89.5
**COVID-19 vaccination situation**		
Have never done before	93	54.1
1 dosage	30	17.4
2 dosages	44	25.6
3 dosages	4	2.3
4 dosages	1	0.6
**Health Evaluation Situation**		
Middle	1	0.6
Good	39	22.7
Very Good	132	76.7

*: 2022 minimum wage value, **: 2022 poverty line.

**Table 2 healthcare-11-02517-t002:** Edinburgh Postpartum Depression Scale and Beck Anxiety Scale values.

	EPDS Grade		BAI Grade		
m ± sd	1.27 ± 3.72		3.85 ± 2.62		
	<13	≥13	˂20	20–30	≥31
Number (n)	167	5	172	-	-
Percentage (%)	97.1	2.9	100	-	-

**Table 3 healthcare-11-02517-t003:** Edinburgh Postpartum Depression Scale and influencing factors.

	EPDS Grade		Pearson Ki-Kare	df	*p*
	0–12	13–30			
**Income Level**			16.246	3	0.001
0–5.324 * TL	150	4			
5.324–17.340 ** TL	10	-			
Not known	6	-			
Other	1	1			
**COVID-19 Vaccination Situation**			14.718	4	0.005
1 Dosage	27	3			
2 Dosage	44	-			
3 Dosage	3	1			
4 Dosage	1	-			
Have never done before	92	1			

*: 2022 minimum wage value, **: 2022 poverty line.

**Table 4 healthcare-11-02517-t004:** Beck Anxiety Scale and Influencing Factors.

	BAI Grade			F	df	*p*
	˂20	20–30	31–63			
**Education level**				3.402	3	0.019
Under primary education ^1^	14	-	-			1–2
Primary education ^2^	107 (62.5%)	-	-			1–3
High school ^3^	43 (25%)	-	-			2–4
Associate-Bachelor ^4^	8	-	-			3–4

F: One-way ANOVA, SD: degrees of freedom, *p*: significance level. ^1,2,3,4^ represent groups categorized based on education levels. The *p*-value indicates which groups have statistically significant differences between them.

## Data Availability

All datasets and analyses used throughout the study are available from the corresponding author upon reasonable request.

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
