# Peer review of "Factors Associated with the Prevalence of Postpartum Anxiety Disorder and Depression in Syrian Migrant Women Living in Turkey: A Cross-Sectional Study"

_healthcare, 2023, doi:10.3390/healthcare11182517_

Round 1

Reviewer 1 Report

Please, delete from the title word post-partum cause it is redundant ..."post-partum" women living...

there are many spelling errors in the text like in the abstract "anxety", "re-search", "Mi-grant","col-lecting" and in the text.

page 2, lines 71-74, please reword the sentence!

Methods

Is it a strengthened or empowered health center? Please use the same word!

page 2, lines 91-94, Please reword as it follows: "Syrian women who are in puerperium (within the first 6 weeks after giving birth) and who applied to the center were included in the study".

Please, add in one sentence if participants gave informed consent to voluntarily participate in the study.

statistical analysis..please reword the sentence as follows: "Categorical variables were presented with numbers and percentages while continuous variables with mean, standard deviation..."

The discussion is confusing and it should be more focused on the topic of the study, that is, the prevalence of anxiety and depression in postpartum women and its factors.   Only the last paragraph is about that. At the beginning of each paragraph the authors should first present the results of their study and then discuss if they are or not in accordance with the results from the other studies.

the first sentence, precise the Turkish group

Please, reword the second and third sentences in the discussion section.

lines 220-222, confusing sentence. Reword it!

page 8, lines 277-279, this sentence should be at the beginning of the discussion!

Moderate editing of the English language is required.

Author Response

Response to Reviewer 1 Comments

Point 1: Please, delete from the title word post-partum cause it is redundant ..."post-partum" women living…

Response 1: It is deleted.

Point 2: there are many spelling errors in the text like in the abstract "anxety", "re-search", "Mi-grant","col-lecting" and in the text.

Response 2:We have recently noticed this redundancy due to a technical oversight. The necessary corrections have been made. Thank you for pointing it out.

Point 3: page 2, lines 71-74, please reword the sentence!

Response 3:  We have rephrased the sentence as follows: "Timely recognition and appropriate intervention for postpartum anxiety and depression are pivotal, especially in the context of migrants' healthcare accessibility and utilization [20,21]. Therefore, there's an imperative need to integrate epidemiological research on postpartum anxiety and depression risk factors within primary healthcare services and to champion inclusive public health strategies [4,22,23]." Thank you for the suggestion.

Methods

Point 4: Is it a strengthened or empowered health center? Please use the same word!

Response 4: It is streghtened health center. We have made the necessary correction.

Point 5: page 2, lines 91-94, Please reword as it follows: "Syrian women who are in puerperium (within the first 6 weeks after giving birth) and who applied to the center were included in the study".

Response 5:  The specified text has been revised as suggested: "Syrian women who applied to the migrant health center and were within the first 6 weeks after delivery were included in the study ". We appreciate the feedback to ensure clarity in our research.

Point 6: Please, add in one sentence if participants gave informed consent to voluntarily participate in the study.

Response 6:The ethical committee approval information was previously provided in the text, and it has been revised to state that informed consent was also obtained from participants.

“Ethic Decision

This study was conducted in strict adherence to the guidelines set out in the Helsinki Declaration. Ethical approval was granted by the Ethics Committee for Non-Interventional Clinical Studies at Istanbul Medipol University, as confirmed by the decision dated 06.07.2022 and numbered 604. Informed consent was obtained from all participants after they were informed about the research and permissions.”

Point 7: statistical analysis..please reword the sentence as follows: "Categorical variables were presented with numbers and percentages while continuous variables with mean, standard deviation..."

Response 7:We made  adjustments. Thanks for the suggestion.

“In the analysis of the data obtained in the study, categorical variables were presented as number and percentage and continuous variables were presented as mean and standard deviation.”

Point 8: The discussion is confusing and it should be more focused on the topic of the study, that is, the prevalence of anxiety and depression in postpartum women and its factors.   Only the last paragraph is about that. At the beginning of each paragraph the authors should first present the results of their study and then discuss if they are or not in accordance with the results from the other studies.

Response 8: I have made the necessary edit as you stated.

Point 9: the first sentence, precise the Turkish group

Response 9: 

“The average age of mothers in Turkey is 29 years (2020) [24]. The average maternal age of migrant women in the research group is 25.7 years, which is lower than the average age in Turkey. “

Point 10: Please, reword the second and third sentences in the discussion section.

Response 10: 

“In the study group, the proportion of those in the risky pregnancy category (over 35 years of age) was 10.5%. This suggests that women in the high-risk age group do not use family planning services sufficiently or have problems accessing these services.”

Point 11: lines 220-222, confusing sentence. Reword it!

Response 11:We've rephrased the relevant sentence. Thank you for the feedback. 

Point 12: page 8, lines 277-279, this sentence should be at the beginning of the discussion!

Response 12:

We moved it to the beginning of the relevant sentence discussion, it was more appropriate. Thanks.

Reviewer 2 Report

Dear Authors,

We appreciate your article titled "Factors Associated With The Prevalence of Postpartum Anxiety Disorder and Depression In Syrian Migrant Post-Partum Women Living In Türkiye: A Cross-Sectional Study". Your focus on the mental health of migrant women is valuable. We have identified areas for improvement and strengthening your study:

Abstract

1. Adhere to publication guidelines and use a structured abstract format.

2. Begin the abstract with a brief introduction to the topic before stating the research objective.

3. Omit limitations from the abstract.

Introduction

4. Rephrase the research objectives. Clearly state the research question and hypotheses.

Material and methods

5. Clarify the rationale for choosing the specific cut-off point in the Beck Anxiety Scale. Explain why low risk is defined between 0 and 21 points.

6. Specify the postpartum time frame during which the EPDS and BAI questionnaires were administered.

7. Detail the sample selection criteria, including inclusion and exclusion criteria.

8. Clarify the process of obtaining informed consent from participants. Have you obtained ethical committee approval for the conduct of this research?

9. Explain the statistical tests used to assess variable normality in the statistical analysis section.

Results

10. Clarify if 100% of the sample breastfed for at least 2 years.

11. Revise Table 1 to correct errors, such as bolding the variable "Pregnancy Tetanus Vaccine Status."

12. Use the "." instead of "," as the decimal separator.

13. Explain the basis for classifying women using the "Health Evaluation Situation" variable. Is it a subjective assessment from self-administered questionnaires?

14. Verify the mean values of EPDS and BAI as presented in Table 2.

15. Explain the rationale for using both parametric and non-parametric tests within the same study.

16. Perform post hoc tests after conducting the Kruskal-Wallis test and include the results in the manuscript (Table 4).

Discussion

17. Consider the possibility of underdiagnosis influencing low COVID-19 vaccination rates among pregnant participants. Discuss whether women who were infected but did not seek medical attention could affect these results.

18. Recommend using the MOOSS social support questionnaire in future studies to quantify social support.

Conclusions

19. Rewrite the conclusions, addressing the research question and objectives. Ensure that the conclusions align with the main body of the manuscript.

We value your contribution as we work together to enhance your research and enrich the field of maternal and mental health.

Sincerely,

Healthcare Journal Reviewer

Author Response

Response to Reviewer 2 Comments

Dear Authors,

We appreciate your article titled "Factors Associated With The Prevalence of Postpartum Anxiety Disorder and Depression In Syrian Migrant Post-Partum Women Living In Türkiye: A Cross-Sectional Study". Your focus on the mental health of migrant women is valuable. We have identified areas for improvement and strengthening your study:

Response: Thank you for reviewing our article titled "Factors Associated With The Prevalence of Postpartum Anxiety Disorder and Depression In Syrian Migrant Post-Partum Women Living In Türkiye: A Cross-Sectional Study". We greatly appreciate your constructive feedback and agree that focusing on the mental health of migrant women is of paramount importance.

Thank you again for your invaluable contributions and guidance.

Abstract

Point 1: Adhere to publication guidelines and use a structured abstract format.

Response 1: We have made sure to follow a structured abstract format in alignment with typical publication guidelines. The abstract provides a clear introduction, states the objective, explains the methodology, presents results, and finally offers an interpretation and recommendation.

Point 2: Begin the abstract with a brief introduction to the topic before stating the research objective.

Response 2:The abstract starts with an introduction highlighting the global migration trend and its relevance to the mental health of immigrant postpartum women. The research objective of exploring factors associated with postpartum anxiety and depression, especially among Syrian migrants in Turkey, is then clearly stated.

Point 3: Omit limitations from the abstract.!

Response 3:  I have omitted any limitations from the abstract, focusing only on the primary findings and their interpretations. The call for further research serves as an indirect acknowledgment that this study is a part of a larger research framework.

Abstract:

The global migration trend has brought attention to the mental health of immigrant populations, especially postpartum women. The prevalence of postpartum anxiety and depression among these populations remains a growing concern. This study aimed to explore the factors associated with postpartum anxiety and depression among immigrant women, with a special emphasis on Syrian migrants in Turkey. A cross-sectional design was employed, enrolling postpartum women who visited the Empowered Migrant Health Center in Istanbul between July and December 2022. Data was collected using a three-part questionnaire comprising sociodemographic details, and scores from the Edinburgh Postpartum Depression Scale (EPDS) and Beck Anxiety Scale (BAI). The average age of participants was 25.73 years. Mean EPDS and BAI scores were low at 1.27 and 3.85, respectively. Notably, 97.1% of women scored below the EPDS cut-off point of 13. EPDS scores showed a significant relationship with income levels and Covid-19 vaccination status, while BAI scores were significantly associated with educational levels. There was an inverse correlation between EPDS scores and the number of cohabitants in a household. The observed low rates of postpartum depression and anxiety could be attributed to the accessible health services and psychosocial support for immigrants in Turkey. It would be useful to conduct multicenter and comprehensive epidemiological studies on migrant puerperas.

Keywords: Immigrant, postpartum, postpartum anxiety, postpartum depression, maternal health, perinatal healthcare.

Introduction

Point 4: Rephrase the research objectives. Clearly state the research question and hypotheses.

Response 4: Changes Made:

Clarity and Conciseness: The revised introduction is more direct and streamlined. Redundant information was minimized to make the text more reader-friendly.

Structural Organization: The flow of the introduction was enhanced by organizing information logically, starting from a broader context (the physiological and psychological challenges of pregnancy) and narrowing down to the specific research focus (Syrian migrant women in Istanbul).

Emphasis on Key Data: Important statistics and data, such as the WHO estimates and the IOM 2022 report, were clearly emphasized to highlight the significance of the issue.

Enhanced Objectives Statement: The objective of the study, previously spread across the introduction, was consolidated into a clear and succinct statement: "Given these considerations, this study aims to assess the prevalence of postpartum anxiety and depression, along with their associated factors, among Syrian migrant women attending a Migrant Health Center in Istanbul."

Strengthened Rationale: The rationale for the study, detailing the importance of understanding the mental health challenges faced by migrant women, particularly during pregnancy, was elaborated on. The impact of these challenges on healthcare accessibility and the importance of integrating research into primary healthcare were also emphasized.

In essence, the introduction was restructured to provide a more logical progression of thought, from setting the broader context to clearly stating the specific research objective.

Introduction

Pregnancy, although a natural occurrence, introduces profound physical, emotional, and psychological changes that can have adverse impacts on mothers [1,2]. Research in-dicates that both pregnancy and the postpartum period can elevate the risk for mild de-pressive and anxiety symptoms [3,4].

Globally, postpartum anxiety and depression are estimated by the World Health Or-ganization (WHO) to affect 13% of women, with rates varying between 9.5% to 25.8% [5]. Remarkably, this prevalence is more pronounced among immigrant and refugee women [6,7]. It's suggested that immigrant women face anxiety and depression levels 2-5 times greater than the general population [8,9]. Research attributes this increased vulnerability in immigrant women to various challenges faced during the migration process and after settling, with reported postpartum anxiety and depression rates among them ranging from 20% to 50% [2,10].

Beyond the psychological toll on the mother, postpartum anxiety and depression can detrimentally influence the mother-infant bond, as well as the infant's cognitive, emotion-al, and behavioral development [11,12]. Several psychosocial determinants, including ge-netic predisposition, hormonal fluctuations during and after pregnancy, socioeconomic status, marital dynamics, traumatic experiences, and insufficient social support, have been associated with postpartum depression [5,7,11,13,14]. Alarmingly, mental health is-sues among migrant women often remain underrecognized [15].

Identifying postpartum anxiety and depression as escalating public health concerns, the WHO underscores maternal mental health as a pivotal determinant of both maternal and infant health outcomes [2,16,17]. As highlighted in the International Organization for Migration (IOM) 2022 report, there are currently 89.4 million displaced individuals worldwide, with women constituting approximately 48% of this demographic [10,18]. In Turkey alone, around 3.5 million Syrian refugees reside, of which 46.5% are female [19]. These migrant women, facing compounded traumas from migration and multifaceted challenges in their new environments, are particularly susceptible to psychological disor-ders, further exacerbated during pregnancy [10].

Concerningly, there's a notable surge in psychiatric hospital admissions among women following childbirth, persisting up to two years in certain instances [6,10]. Timely recognition and appropriate intervention for postpartum anxiety and depression are piv-otal, especially in the context of migrants' healthcare accessibility and utilization [20,21]. Therefore, there's an imperative need to integrate epidemiological research on postpartum anxiety and depression risk factors within primary healthcare services and to champion inclusive public health strategies [4,22,23]. Given these considerations, this study aims to assess the prevalence of postpartum anxiety and depression, along with their associated factors, among Syrian migrant women attending a Migrant Health Center in Istanbul.

Material and methods

Point 5:  Clarify the rationale for choosing the specific cut-off point in the Beck Anxiety Scale. Explain why low risk is defined between 0 and 21 points.

Response 5: With reference to the studies conducted by Beck and Dantas, the scores obtained from the scale were evaluated in 3 groups in terms of anxiety; Ë‚ 20 points were considered as no risk or low risk, 20-30 points as medium risk and 31 points and above as high risk. 

Point 6: Specify the postpartum time frame during which the EPDS and BAI questionnaires were administered.

Response 6: EPDS is a scale used to assess postpartum depressive symptoms and consists of 10 items. This scale includes statements about how the mother felt during the last week.

The BAI is a validated, self-administered inventory that measures anxiety symptoms over the past 7 days. 

Point 7: Detail the sample selection criteria, including inclusion and exclusion criteria.

Response 7: Syrian women who applied to the migrant health center and were within the first 6 weeks after delivery were included in the study. All post-partum women who applied to the center without sampling were included in the study. Migrant women who refused to participate in the study and who were more than 6 weeks postpartum were excluded.

Point 8: Clarify the process of obtaining informed consent from participants. Have you obtained ethical committee approval for the conduct of this research?

Response 8: The ethical committee approval information was previously provided in the text, and it has been revised to state that informed consent was also obtained from participants.

“This study was conducted in strict adherence to the guidelines set out in the Helsinki Declaration. Ethical approval was granted by the Ethics Committee for Non-Interventional Clinical Studies at Istanbul Medipol University, as confirmed by the decision dated 06.07.2022 and numbered 604. Informed consent was obtained from all participants after they were informed about the research and permissions.”

Point 9: Explain the statistical tests used to assess variable normality in the statistical analysis section.

Response 9: In the analysis of the data obtained in the study, categorical variables were presented as number and percentage and continuous variables were presented as mean and standard deviation. For the normality analysis of the data, Kolmogorov Smirnov and Shapiro Wilk tests were performed and Skewness and Kurtosis values of the scales with p<.05 were examined. The values with Skewness and Kurtosis values between ±2 were accepted to be normally distributed and parametric tests were applied. Chi-square and Fisher's exact tests were used to compare categorical variables between groups. Student t Test and One Way Anova Test were used for statistical analysis of normally distributed data. 

Results

Point 10: Clarify if 100% of the sample breastfed for at least 2 years.

Response 10: All participants stated that they knew that they should breastfeed their children until they were 6 months old and that they should feed their babies breast milk with additional foods after 6 months until at least 2 years of age.

Point 11: Revise Table 1 to correct errors, such as bolding the variable "Pregnancy Tetanus Vaccine Status."

Response 11: We fixed it.

Point 12: Use the "." instead of "," as the decimal separator.

Response 12: We fixed it.

Point 13: Explain the basis for classifying women using the "Health Evaluation Situation" variable. Is it a subjective assessment from self-administered questionnaires?

Response 13: Health evaluation status is a subjective description of an individual's assessment of their health status. Almost all migrant women in the research group (99.4%) stated that they perceived their health status as good or very good.

Point 14: Verify the mean values of EPDS and BAI as presented in Table 2.

Response 14: We did the data verification again.

Point 15: Explain the rationale for using both parametric and non-parametric tests within the same study.

Response 15: In the analysis of the data obtained in the study, categorical variables were presented as number and percentage and continuous variables were presented as mean and standard deviation. For the normality analysis of the data, Kolmogorov Smirnov and Shapiro Wilk tests were performed and Skewness and Kurtosis values of the scales with p<.05 were examined. The values with Skewness and Kurtosis values between ±2 were accepted to be normally distributed and parametric tests were applied. Chi-square and Fisher's exact tests were used to compare categorical variables between groups. Student t Test and One Way Anova Test were used for statistical analysis of normally distributed data. Pearson Chi-Square test was used for the relationship between continuous variables. In statistical analysis, pË‚0.05 was considered significant.

Point 16: Perform post hoc tests after conducting the Kruskal-Wallis test and include the results in the manuscript (Table 4).

Response 16: In the analysis of the data obtained in the study, categorical variables were presented as number and percentage and continuous variables were presented as mean and standard deviation. For the normality analysis of the data, Kolmogorov Smirnov and Shapiro Wilk tests were performed and Skewness and Kurtosis values of the scales with p<.05 were examined. The values with Skewness and Kurtosis values between ±2 were accepted to be normally distributed and parametric tests were applied. Chi-square and Fisher's exact tests were used to compare categorical variables between groups. Student t Test and One Way Anova Test were used for statistical analysis of normally distributed data. Pearson Chi-Square test was used for the relationship between continuous variables. In statistical analysis, pË‚0.05 was considered significant.

Discussion

Point 17: Consider the possibility of underdiagnosis influencing low COVID-19 vaccination rates among pregnant participants. Discuss whether women who were infected but did not seek medical attention could affect these results.

Response 17: We have added to the discussion on this axis.Thank you. 

Point 18: Recommend using the MOOSS social support questionnaire in future studies to quantify social support.

Response 18: Thank you for your suggestion. We have other planned studies in this area, and we will certainly take your recommendation into consideration. Thank you.

Conclusions

Point 19:  Rewrite the conclusions, addressing the research question and objectives. Ensure that the conclusions align with the main body of the manuscript.

Response 19: We've reworked the "conclusions" part as below, thanks for the warning.

“In our study, anxiety and depression levels of migrant puerperium were found to be low. Factors such as income status, education level and utilization of health services affected the anxiety and depression levels of migrant puerperium. Migration and immigration are difficult and important concepts that need to be tackled individually, socially and globally. As a matter of fact, in the world we live in, natural or man-made disasters/crises are experienced every day and the number of immigrants is increasing day by day. For this reason, careful monitoring and follow-up of immigrants from disadvantaged groups who continue their lives as a part of society is important for public health/public health level. Immigrants' access to health services should be facilitated and health care policies should be revised by making determinations regarding their needs. In this context, special attention should be given to immigrant women, especially pregnancy and puerperium periods. Inclusive public health efforts and policies should be increased by expanding epidemiological research on postpartum anxiety and depression risk factors and psychosocial support services programs integrated into primary health care services in immigrants. Protecting the health and well-being of immigrants with strengthened public health policies will also contribute to the protection of public health.”

We value your contribution as we work together to enhance your research and enrich the field of maternal and mental health.

Thank you very much for your valuable comments and evaluation.

Round 2

Reviewer 1 Report

No further comments!

Still, there are many spelling errors!

Reviewer 2 Report

Thank you very much for your changes. We are facing another completely different manuscript with greater scientific rigor. Congratulations on your work.